# Syntax Error-Free and Generalizable Tool Use for LLMs via Finite-State Decoding

## Abstract

*For reviewers' convenience, we highlight the revised parts like this sentence.*
Large language models (LLMs) have shown promising capabilities in using external tools to solve complex problems. However, existing approaches involve either fine-tuning on tool demonstrations, which does not generalize to new tools without additional training, or providing tool documentation in context, limiting the number of tools. Both approaches often generate syntactically invalid tool calls. In this paper, we propose TOOLDEC, a finite-state machine-guided decoding algorithm for tool-augmented LLMs. TOOLDEC eliminates tool-related errors for any tool-augmented LLMs by ensuring valid tool names and type-conforming arguments. Furthermore, TOOLDEC enables LLM to effectively select tools using only the information contained in their names, with no need for fine-tuning or in-context documentation. We evaluated multiple prior methods and their TOOLDEC-enhanced versions on a variety of tasks involving tools like math functions, knowledge graph relations, and complex real-world RESTful APIs. Our experiments show that TOOLDEC reduces syntactic errors to zero, consequently achieving significantly better performance and as much as a 2x speedup. We also show that TOOLDEC achieves superior generalization performance on unseen tools, performing up to 8x better than the baselines.

## 1 Introduction

Augmenting large language models (LLMs) with external tools (Mialon et al., 2023) enables them to solve complex problems. Current LLMs can utilize retrievers (Shen et al., 2023; Gupta & Kembhavi, 2022; Schick et al., 2023), RESTful APIs (Qin et al., 2023; Song et al., 2023), program interpreters (Chen et al., 2022; Gao et al., 2023), and various other tools. The performance of a tool-augmented LLM depends on three key decisions—when to use a tool, which tool to use, and how to invoke a tool. Existing approaches learn to make these decisions through fine-tuning or in-context learning.

However, these approaches still generate erroneous tool calls. For example, in-context learning can easily generate non-existent tool names that are not in the tool inventory because non-existent tools may also look plausible as the next token (Song et al., 2023; Qin et al., 2023). Fine-tuned models, though usually call tools by correct names, often pass invalid arguments to the right tool functions (Hao et al., 2023), just as in-context learning does. Furthermore, prior approaches do not generalize to unseen tools well. Fine-tuning approaches need additional training data and further fine-tuning to adopt new tools. In-context learning approaches require tool documentation in the prompts.

To address these issues, we propose TOOLDEC, a decoding algorithm guided by a finite-state machine (FSM) to ensure LLMs invoke tools properly. Our core insight is to explicitly represent states during LLM decoding. Each state is associated with a valid set of tokens corresponding to tool names and tool arguments. TOOLDEC transitions from state to state as decoding progresses. At each decoding step, TOOLDEC samples from a subset of tokens allowed by the current state. The FSM that gives guidance to TOOLDEC is constructed from tool documentation and API signature so that the machine precisely represents the grammar of tool calls. In this way, TOOLDEC is able to always generate syntactically correct tool calls. Figure 1 illustrates that an LLM enhanced by TOOLDEC is able to generate the right function call `multiply` with precise arguments ("3.14" and "123") and therefore, it gets the correct result returned by the tool. More examples comparing TOOLDEC and other tool LLMs can be found in Appendix A.3.

Table 1: The comparison between the LLM tool use paradigms. Finite-state decoding is able to eliminate syntax errors, adopt new tools without new data and in-context documentation.

| Paradigm | Syntax Error-Free | Data-Free Generalization | No Tool Docs in Context |
|---|---|---|---|
| Fine-tuning | ✗ | ✗ | ✓ |
| In-context learning | ✗ | ✓ | ✗ |
| Finite-state decoding (ours) | ✓ | ✓ | ✓ |

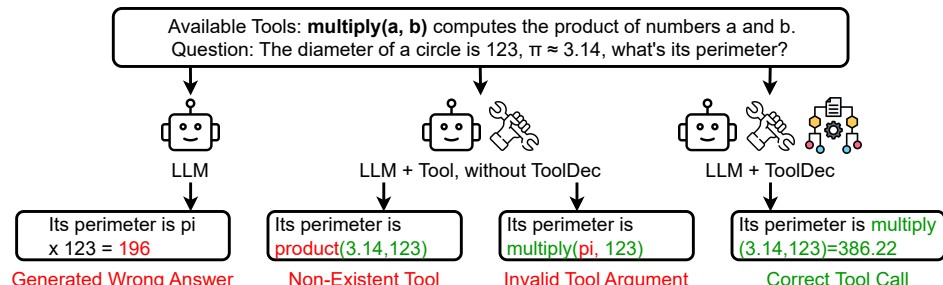

Figure 1: LLMs using external tools. LLMs without tools cannot multiply, so they just generate a probable next token. Tool-augmented LLMs can access external tools like `multiply`, but they may call a non-existent tool like `product` and pass invalid arguments like the string "pi". Our proposed TOOLDEC always generates tool calls without syntax errors.

Furthermore, TOOLDEC generalizes to new tools that never appeared before much more efficiently. Unlike prior approaches which require fine-tuning or in-context descriptions of new tools, TOOLDEC automatically constructs a finite-state machine from a tool's API signature (its name and argument types) and adds it to the existing FSM. TOOLDEC is then able to call new tools without fine-tuning or in-context demonstration. While pre-trained language models can generate tool names when prompted to, they often hallucinate plausible tool names that are not in the inventory. TOOLDEC does not do that. In Figure 1, both `product` and `multiply` sound plausible for the scenario, but only `multiply` is a given tool. Since TOOLDEC only calls existing tools, it won't hallucinate a plausible yet non-existent tool and can rely on the tool names to find the right tool.

The contributions of this paper can be summarized as follows:

- We propose TOOLDEC, a finite-state decoding algorithm to empower LLMs to use tools properly. TOOLDEC enjoys two advantages: its generated tool calls are guaranteed to be syntactically correct and it generalizes to unseen tools efficiently.
- We empirically verify TOOLDEC's superior performance compared to prior strong baselines on four diverse datasets from different domains. Our extensive experiments show TOOLDEC eliminates all syntax errors and hallucinated tool names, resulting in better accuracy and as much as 50% less inference time. Our results also indicate that TOOLDEC is more than 8x better than baselines on mathematical reasoning with 9 unseen tools and 7x better than knowledge question answering with 204 unseen tools.

## 2 RELATED WORK

**Fine-tuning language models to use tools.** Language models can be fine-tuned to use tools with data that contain interleaving text and tool use. Earlier studies make language models use a single tool like a retrieval module (Borgeaud et al., 2022; Guu et al., 2020) or a search engine (Nakano et al., 2021) by fine-tuning. Recent advances in tool-augmented language models that use multiple tools (Schick et al., 2023; Parisi et al., 2022) also fine-tune language models to use tools including QA models, translation models, calculators, and search engines. ToolkenGPT (Hao et al., 2023) proposes to use several special tokens to represent tools and only tunes the embeddings of the tokens so that new tool adoption can be more efficient. However, fine-tuning approaches for tool use still need new data and extra fine-tuning to adapt a model to new tools. We list the differences between finite-state decoding and the previous two paradigms in Table 1.

**In-context learning for tool use.** Language models can learn from in-context examples (Brown et al., 2020) and follow instructions (Ouyang et al., 2022). This makes it possible to simply put the descriptions of tools in the prompt and ask language models to use them. Recent works have used this possibility to use neural models (Shen et al., 2023), RESTful APIs (Qin et al., 2023; Song et al., 2023), program interpreters (Chen et al., 2022; Gao et al., 2023) and many other tools to solve problems. In-context learning does not need extra model tuning to use new tools. However, the description and documentation of new tools still need to be in the prompt, which increases computation cost and limits the context budget for the model to actually reason about the task.

**Constrained decoding and finite-state machines.** Previous constrained decoding methods mainly focus on *lexical* constraints (Anderson et al., 2017). They reduce the large search space of lexically constrained decoding with finite-state machines (Anderson et al., 2017), grouping together similar candidates (Hokamp & Liu, 2017), and better search algorithms (Miao et al., 2019; Lu et al., 2021; 2022). However, lexical constraints are not expressive enough to regulate tool calls. While finite-state machines have to be weighted and probabilistic to deal with the soft constraints in natural language (Eisner, 2002; Rastogi et al., 2016), the constraints for syntactic tool calls are hard constraints that are much easier for FSMs. Therefore, we propose TOOLDEC to meet the *syntactic* constraints of a valid tool call.

# 3  TOOLDEC: LLM TOOL USE VIA FINITE-STATE DECODING

A syntactically correct tool call needs to refer to an existent tool name and pass type-conforming arguments. Motivated by the fact that it is straightforward to verify the syntax of a tool call using a finite-state machine (FSM), we propose TOOLDEC, a constrained decoding algorithm guided by an FSM. During each decoding step, the model samples from a subset of the vocabulary that only contains syntactically correct tokens. The FSM that specifies the token subsets can be automatically constructed from the tool documentation. We describe the implementation details of the automatic construction pipeline in Appendix A.9 For example, in Figure 2, an FSM is constructed for functions `add`, `exp`, `square` and `sqrt`. Table 2 shows how TOOLDEC answers the question "the side of a square is 5, what's its area?" using the FSM. With the guidance from the FSM, TOOLDEC achieves the following goals:

- **Switching Modes.** Switch between "text mode" when the model is free to generate any text and "tool mode" when the model can only generate valid tool calls.
- **Generating Tool Names.** At the beginning of a tool call, only generate *correct existent* tool names from a pre-defined list of tools.
- **Passing Arguments.** Only pass type-conforming arguments to the tool.

## 3.1  FINITE-STATE DECODING

TOOLDEC is guided by a finite-state machine (FSM). An FSM is a 5-tuple $(S, V, g, s_0, R)$, consisting of a finite state set $S$, an alphabet $V$, a transition function $g : S \times V \to S$, an initial state $s_0$ and a set of accepting states $R$. In our case, $S$ and $g$ are constructed from the tool signature. $V$ is the token vocabulary of the language model. $R$ corresponds to pre-defined tokens that can determine the LM has completed the task, like '<EOS>'.

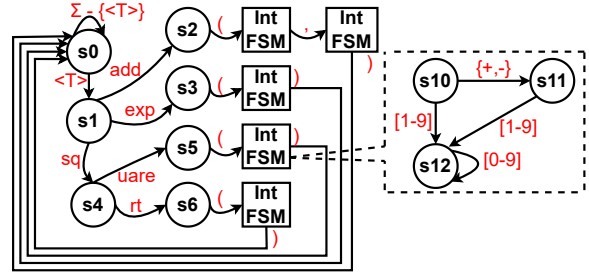

Figure 2: A finite-state machine for TOOLDEC constructed for math functions `add`, `exp`, `square`, `sqrt` that take integers as arguments. The names of the tools are represented with a trie structure. "IntFSM" is a submodule that parses integers.

At each decoding step $t$, TOOLDEC maintains a current state $s$. It can only generate the tokens permitted by the FSM, i.e. the tokens for which $g(s, \cdot)$ is defined. These permitted tokens are a subset of $V$ and we denote them as $V_s$. After generating one token $a$, TOOLDEC transits to another state $g(s, a)$ specified by the FSM transition function. The permitted tokens for each state can be the full vocabulary, or a valid subset corresponding to tool names

Table 2: How TOOLDEC uses the FSM in Figure 2 to answer the question "the side of a square is 5, what's its area?". At each time step, the state in the machine corresponds to a set of valid next tokens. We zero out all other tokens and re-normalize the next token distribution, forcing the LLM to only sample valid tokens.

| Step | Generated Text | Current State $g$ | Valid Next Tokens | Next Token | Next State |
|------|---------------|-------------------|-------------------|-----------|-----------|
| $t$ | Its | $s_0$ | whole vocab. | 'area' | $s_0$ |
| $t+1$ | Its area | $s_0$ | whole vocab. | 'is' | $s_0$ |
| $t+2$ | Its area is | $s_0$ | whole vocab. | '<T>' | $s_1$ |
| $t+3$ | Its area is <T> | $s_1$ | 'add', 'exp', 'sq' | 'sq' | $s_4$ |
| $t+4$ | Its area is <T>sq | $s_4$ | 'uare', 'rt' | 'uare' | $s_5$ |
| $t+5$ | Its area is <T>square | $s_5$ | '(' | '(' | $s_{10}$ |
| $t+6$ | Its area is <T>square( | $s_{10}$ | '+','-','1'-'9' | '5' | $s_{12}$ |
| $t+7$ | Its area is <T>square(5 | $s_{12}$ | '0'-'9', ')' | ')' | $s_{10}$ |
| $t+8$ | Its area is <T>square(5) | $s_{10}$ | whole vocab. | - | - |

and argument types. At each step $t$, we do not directly sample from the next token distribution $P(x_t|x_{1..t-1})$ calculated by the LLM. Instead, we zero out the probabilities of invalid tokens for which the transition function is undefined, and normalize the probabilities,

$$\tilde{P}(x_t = a|x_{1..t-1}, s) = \begin{cases} \frac{P(x_t=a|x_{1..t-1})}{\sum_{a' \in V_s} P(x_t=a'|x_{1..t-1})}, & g(s,a) \text{ is defined,} \\ 0, & \text{otherwise} \end{cases}.$$

The next token $a$ is then sampled from the modified distribution $\tilde{P}(x_t|x_{1..t-1}, s)$. With the next token, we move on to the next decoding step and transition the current state $s$ to the next state $g(s,a)$. The pseudo-code of this algorithm is listed in Appendix A.1.

## 3.2 CONSTRUCTING FSMS THAT GUARANTEE SYNTACTICALLY CORRECT TOOL CALLS

**Switching Modes.** We use two states in the FSM to represent whether the language model is in text mode or tool mode. The text mode is denoted by the initial state $s_0$, during which the model is free to generate any token in its vocabulary, i.e. $V_{s_0} = V$. The tool mode is denoted by state $s_1$. The model needs to output the special token <T> to switch from text mode to tool mode. Starting at $s_1$ is a smaller FSM that describes the grammar of tool calls. Note that TOOLDEC can be applied to models with different mode-switching schemes. For some models like ToolkenGPT (Hao et al., 2023) and toolformer (Schick et al., 2023), when to use a tool is decided by the model itself using special tokens, just like in Table 2. For others like ToolLLM (Qin et al., 2023), the planning is done explicitly using an external planner similar to that in ReAct (Yao et al., 2023). For those models, TOOLDEC switches between $s_0$ and $s_1$ according to the external planner.

**Generating Tool Names.** Once the model is at state $s_1$, the next step would be to generate a new tool call. To generate a correct tool call, the model needs to output the correct name for the tool and a correct list of arguments. Many tools have long names that need multiple tokens to represent. For example, the tool `square` in Figure 2 consists of two tokens— 'sq' and 'uare'. Therefore, we need to construct an FSM for multi-token tool names.

We use a trie (Fredkin, 1960) to construct the FSM for tool names. In Figure 2, the trie for tool names consists of states $s_1, \ldots, s_6$. A trie is a rooted tree in which each edge represents a token. A tree node represents a string that's the concatenation of the path from the root to this node. In our example, $s_4$ represents "sq" and $s_5$ represents "square". To construct a trie, we insert all the strings into it one by one. Inserting a string into a trie means going from the root down the path made by the string and creating new nodes when the next step in the path does not exist. For example, we show how two more tools names, `exp10` and `expand` can be added to the trie in Figure 3.

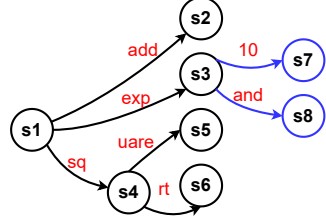

Figure 3: Adding two more tools `exp-10` and `expand` into the trie, represented by the two blue states $s_7$ and $s_8$.

**Generating Syntactically Valid Tool Arguments.** Tool arguments have specified types. Like arguments in a program, they need to follow certain grammar rules. These rules can be specified by finite-state machines. For example, the "IntFSM" in Figure 2 depicts a finite-state machine that only accepts integer literals. For all arguments of a tool, we chain their corresponding FSMs together and use the last state corresponding to the tool name as the initial state of this FSM chain. Note that in practice, it's not necessary to explicitly construct this FSM. Any grammar checker that tells the set of valid next tokens suffice.

## 3.3 GENERALIZING TO UNSEEN TOOLS

**Why TOOLDEC generalizes to new tools.** The generalization ability of TOOLDEC is based on two assumptions: i) LLMs already know the plausible-sounding tool names in different scenarios; ii) the tool names in the tool inventory are meaningful and indicative of their usage. For example, we can describe one tool-related scenario with the prompt "*I would like to compute the area of a square with the tool*". In this scenario, the first assumption guarantees that "*power*", "*multiply*", "*times*", "*square*" are among the likely next tokens predicted by the LLM. The second assumption guarantees that the actual name of the tool in our inventory looks like "*power*", "*multiply*", "*times*", "*square*", instead of random strings such as "*jeboglpspr*". With these assumptions, TOOLDEC can select the existent tool names from the plausible-sounding tool names predicted by the LLM.

**What if the second assumption does not hold?** The first assumption is dependent on the ability of the underlying LLM. The second assumption is reasonable because of the naming convention of variables well established in the software engineering industry (Weinberger et al., 2013) and open-source community (Torvalds et al., 2001).

Even when the tool functions are not named or poorly named, LLMs themselves can be utilized to create semantically meaningful function names. In this case, we can prompt LLMs with the documentation or description of the tool's functionality and the heuristics for meaningful tool names. The prompts we used are listed in Appendix A.6.

## 4 EXPERIMENT I: TOOLDEC ELIMINATES SYNTAX ERRORS

In this section, we show that TOOLDEC can eliminate syntax errors while generating tool calls. We select two recent baselines, ToolLLM and ToolkenGPT, representative of the in-context learning and the fine-tuning paradigm, to showcase TOOLDEC's ability. Since the tool-use settings for the two baselines are different and cannot be applied to one another, we test TOOLDEC's performance separately for the two baselines using the benchmarks from the original papers. Through extensive experiments, we show that TOOLDEC can completely eliminate syntactic errors, resulting in better accuracy and shorter inference time.

### 4.1 BASELINES AND BENCHMARKS

**ToolLLM (Qin et al., 2023).** ToolLLM is an in-context learning approachIt utilizes an instruction-tuned LLaMA-7B model (Touvron et al., 2023) to use tools. Given the natural language instruction of a tool-dependent task, an API retriever first retrieves a small subset of relevant functions. Textual description and schema of these relevant functions are made available in the context. Then, ToolLLM goes through a multi-step reasoning process using the functions to produce a final answer.

ToolLLM is evaluated on **ToolEval**, a dataset proposed in the same paper. ToolEval contains tasks that involve a massive set (10,000+) of publicly available REST APIs. We use the most difficult subsets of ToolEval to evaluate our method—I2-Category and I3-Instruction. They contain tasks that need complex and unseen tools from multiple categories (such as geolocation, date/time, and others) to solve. On average, a task in these subsets need more than 7 tools to solve. ToolEval has two main metrics: **Pass Rate** measures the percentage of tasks for which the model reaches an answer within limited reasoning steps. **Win Rate** utilizes an automatic evaluator powered by an LLM following a pre-defined set of criteria for a better path. It compares the quality and correctness of the models' answers to the reference answers from ChatGPT. Qin et al. (2023) finds that the

automatic evaluator has a high correlation of 75.8% with human annotators. Other than these two metrics, we also measure **Tool Error Rate**, the proportion of tasks that have at least one tool-related error.

**ToolkenGPT (Hao et al., 2023).** ToolkenGPT is a fine-tuning approach to tool use. ToolkenGPT represents each tool as a special token and optimizes only the embedding of the tool tokens for tool use. During inference, ToolkenGPT invokes a tool once the corresponding special token is predicted. During a tool call, it passes arguments by learning from in-context demonstrations. ToolkenGPT uses LLaMA-33B (Touvron et al., 2023) as its base model.

We evaluated TOOLDEC's performance on **FuncQA$_{multi}$**(Hao et al., 2023). This dataset tests LLMs' ability in numerical reasoning tasks with 68 math problems. Then, LLMs are required to produce a numerical answer using a few of the 13 arithmetic operations as tools (e.g. `multiply`, `power`, `lcm`). The accuracy is determined by measuring the percentage of problems for which a correct answer is produced, with a 0.1% error tolerance. On average, a problem in FuncQA$_{multi}$ requires 2.78 tool calls to solve. Following Hao et al. (2023), we report results of other baselines, including ChatGPT without tools, LLaMA with chain-of-thought and tools, LLaMA with ReAct and tools.

## 4.2 INTEGRATING TOOLDEC WITH THE BASE MODELS

**ToolLLM+TOOLDEC.** Following Qin et al. (2023), we use ReAct (Yao et al., 2023) to plan the tool calls of ToolLLM. This conforms to the second case of mode switching in Section 3.2. There are three parts in the FSM for ToolLLM. First, a format FSM that enforces the "Thought, Action, Action Input" syntax of ReAct. After decoding "Action:", this FSM transitions to the beginning state of the function name FSM, which guarantees that a decoded function name is always valid. We also constructed a JSON-based function argument FSM. We allowed LLMs to reason for 5 steps before it must call the finish action to be considered "pass".

**ToolkenGPT+TOOLDEC.** Since ToolkenGPT uses special tokens to call tools, TOOLDEC is only applied to guarantee the syntax of arguments. In this experiment, our FSM guarantees that every argument is a valid number, and arguments are separated by commas. It also guarantees that the actual number of arguments passed to a function is exactly the number needed by it. We compared TOOLDEC to two variants of the baseline in Hao et al. (2023), one with backtrace and one without. Backtrace tries to avoid failed tool calls by allowing the LLM to go back and try the next probable token, in place of the failed tool call. To evaluate TOOLDEC, we report the average inference time per problem and tool error rates in addition to accuracy.

## 4.3 EXPERIMENTAL RESULTS

**TOOLDEC enhances in-context learning tool LLMs.** Table 3 shows TOOLDEC's performance on ToolEval. TOOLDEC achieved 55% win rate on I2-Category and 60% win rate on I3-instruction. As a drop-in replacement of the original decoding algorithm, TOOLDEC eliminated all three types of tool-related errors and achieved the best win rate and pass rate, even beating ChatGPT.

The high tool error rate of the baselines suggests that even after instruction fine-tuning, ToolLLM still lacks the ability of accurately invoke external tools from the tool documentation. This inability is more exposed when there is a large variety of tools available, as in I3-Instruction. Furthermore, these errors significantly impacted the model's ability to complete tasks.

Table 3: Results on ToolEval. TOOLDEC-enhanced ToolLLM outperformed baselines ToolLLM in all metrics. TOOLDEC eliminated all tool errors and was even able to beat ChatGPT slightly.

|  | I2-Category | | | I3-Instruction | | |
|---|---|---|---|---|---|---|
|  | Pass ↑ | Win ↑ | Tool Error ↓ | Pass ↑ | Win ↑ | Tool Error ↓ |
| ChatGPT | 39.0% | - | 8.0% | 29.0% | - | 7.0% |
| ToolLLM | 32.0% | 44.0% | 39.0% | 18.0% | 44.0% | 47.0% |
| ToolLLM + Fuzzy Matching | 38.0% | 48.0% | 19.0% | 23.0% | 49.0% | 23.0% |
| ToolLLM + TOOLDEC (**ours**) | **42.0%** | **55.0%** | **0.0%** | **34.0%** | **60.0%** | **0.0%** |

We present the error rates of each error type on two benchmarks in Figure 4. For ToolLLMs, name error, i.e. calling a non-existent tool, was the most common syntactic error in a tool call. TOOLDEC completely eliminated all three errors.

With function name hallucination being the most prevalent tool-related error, a better baseline is to mitigate it with fuzzy matching by suffix. We present the results of the baseline with fuzzy matching as ToolLLM + Fuzzy Matching, and without as ToolLLM. This mitigation increased pass rate but had little impact on win rate, as evident in Table 3, because wrong APIs could often be chosen when a model was unable to precisely call the wanted tool. Overall, our experiments on ToolLLM demonstrate TOOLDEC is highly effective on in-context learning LLMs. Through the next baseline, ToolkenGPT, we show that TOOLDEC is also beneficial to fine-tuned tool LLMs.

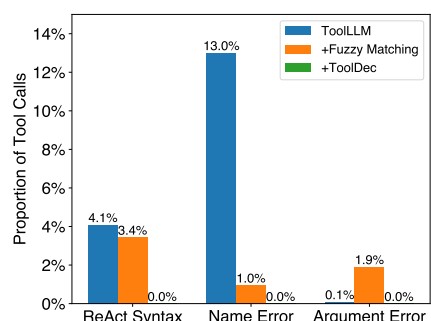

Figure 4: Error rates of three types of tool-related errors relative to the total number of tool calls. TOOLDEC reduced all tool-related errors to *zero*.

**TOOLDEC enhances fine-tuning tool LLMs.** Table 4 shows the results on FuncQA$_{multi}$. We measure the clock inference time of TookenGPT baselines and our method using the same model checkpoint, same environment, and same codebase. Although ToolkenGPT eliminates the possibility of calling non-existent tool names by fine-tuning a special token embedding, it can still suffer from other syntactic errors, which is demonstrated by the 27.9% tool error rate. As a drop-in replacement, TOOLDEC increased ToolkenGPT's accuracy while being much faster in inference. Although ToolkenGPT + backtrace achieved slightly better accuracy than TOOLDEC, it used 2x more time to try different tools. TOOLDEC is much faster than ToolkenGPT baselines because the baselines waste a lot of time for the erroneous tool calls and retrying, while TOOLDEC has no syntactic errors. We show some examples that demonstrate this phenomenon in Appendix A.5. Note that since TOOLDEC eliminated all tool errors, there were no failed tool calls for backtrace to retry. The results underscore tool-related errors' relevancy and TOOLDEC's applicability to both recent in-context learning and fine-tuning tool-augmented LLMs.

Table 4: Results on FuncQA$_{multi}$. ToolkenGPT combined with TOOLDEC eliminated all tool errors. Compared to ToolkenGPT without backtrace, TOOLDEC was better in both accuracy and inference time. Compared to ToolkenGPT with backtrace, TOOLDEC achieved a comparable accuracy with only half the time.

|  | Accuracy ↑ | Time ↓ | Tool Error ↓ |
|---|---|---|---|
| 0-shot ChatGPT w/o tools | 9% | - | - |
| LLaMA w/ tools + CoT | 3% | - | - |
| LLaMA w/ tools + ReAct | 6% | - | - |
| ToolkenGPT | 10.3% | 7.76s | 27.9% |
| ToolkenGPT + Backtrace | **14.7%** | 10.39s | 0.0% |
| ToolkenGPT + TOOLDEC (ours) | 13.2% | **5.95s** | **0.0%** |

## 5 EXPERIMENT II: TOOLDEC ENABLES GENERALIZABLE TOOL SELECTION

In Experiment II, we show how TOOLDEC generalizes to unseen tools without additional training data and tool documentation. We compare TOOLDEC with two strong baselines—ToolkenGPT (Hao et al., 2023) as a representative fine-tuning approach, and RestGPT (Song et al., 2023) as a representative in-context learning approach. We conducted experiments on three benchmarks—FuncQA (Hao et al., 2023) and KAMEL (Kalo & Fichtel, 2022), and RestBench (Song et al., 2023). These benchmarks require diverse tool sets from very different domains, including math reasoning, knowledge question answering and real-world web services. In all three domains, our results show that TOOLDEC is able to efficiently generalize to new tools without fine-tuning on extra data.

## 5.1 FINE-TUNING BASELINE: TOOLKENGPT

ToolkenGPT is a fine-tuning approach to tool use that learns a special token for every tool. To generalize to new tools, ToolkenGPT still needs additional data and extra fine-tuning involving the use of new tools. We demonstrate that TOOLDEC, once fine-tuned on a given set of seen tools, doesn't need additional data and further fine-tuning to adopt unseen tools. We compare TOOLDEC and the baselines by tuning them on the same subset of tools, denoted as "seen tools", and then evaluate their performance on "unseen tools".

To guarantee a fair comparison, We mimic ToolkenGPT's planning method to solve the "when to use tools" problem. Specifically, we fine-tune the embedding of a single special token `<T>` to represent all tools, reducing the size of extra vocabulary to 1. Once `<T>` is generated, a tool call begins. We prompt LLM to generate a tool name. The generation of this tool name is guided by an FSM constructed from a list of all available tools. This tool name is then plugged back into the context to start the generation of arguments. We show an example of this process in Appendix A.2.

We selected a small subset of "seen tools" out of all available tools and tuned the embedding only with demonstrations of the tools in the selected subset. We fine-tuned the baseline using the same subset. Then, we evaluated our method and the baselines on tasks that involve unseen tools in the subset to demonstrate TOOLDEC's generalization ability.

**Benchmark on Math Functions**. We use FuncQA multi-hop questions to evaluate our method. Tools in FuncQA, such as `permutate`, `gcd`, `power` are mathematical functions that strictly limits on their arguments to be numbers in certain ranges. We select 4 out of 13 tools as the seen subset to tune the baseline and evaluate different approaches on the remaining 9 unseen tools.

**Benchmark on Knowledge Graph Relations.** To further investigate TOOLDEC's generalizability on a larger set of tools, we also evaluate on KAMEL (Kalo & Fichtel, 2022), a question-answering dataset containing a total of 234 knowledge relations that resemble the characteristics of APIs (e.g. `number_of_children`). More examples can be found in Appendix A.4. The tools in KAMEL are many more than those in FuncQA. They are also more complex and diverse because their number of arguments varies from 1 to 3, and their types include strings, locations, dates, numbers and other ad-hoc types. We select 30 out of 234 tools as the seen subset and evaluate on 4 different evaluation sets, with 30, 60, 100, and 234 tools, respectively. Following Hao et al. (2023), we use prompting, few-shot, and zero-shot as extra baselines. Prompting relies on the internal knowledge of LLM, since no tool was provided. Few-shot demonstrates tool use through few-shot examples. Zero-shot provides descriptions of available tools in context. We make sure that the baselines and TOOLDEC share the same set of seen tools. We use the accuracy of tool calls as a metric, which is determined by the proportion of responses that invoke the correct knowledge relation.

## 5.2 IN-CONTEXT LEARNING BASELINE: RESTGPT

RestGPT (Song et al., 2023) is an in-context learning approach that learns tool use from in-context tool documentation. We demonstrate the generalization ability of TOOLDEC-enhanced RestGPT by showing that RestGPT with TOOLDEC can achieve better accuracy without in-context documentation than the RestGPT baseline with documentation. Since TOOLDEC needs access to the next token distribution, we use Vicuna-based (Zheng et al., 2023) RestGPT as the baseline. For our method, we remove all tool documentation from the prompt, leaving only the instructions for reasoning.

**Benchmark on APIs for Real-World Web Services.** We evaluate on RestBench (Song et al., 2023). It consists of tasks in real-world scenarios including TMDB, a website for movie information, and Spotify, an online music player. These tasks directly come from real-user instructions and require multiple tools in the form of RESTful APIs to solve. RESTful APIs are the *de facto* standard for web services (Li et al., 2016) that use HTTP methods to manipulate resources. The ground truth solutions are annotated by humans in the form of tool call chains. We evaluate our method and the baseline on TMDB, which consists of 55 RESTful APIs. Since HTTP methods such as `GET` and `POST` do not have explicit tool names, we use an LLM to rewrite these APIs to follow this format. The prompts we used are in Appendix A.6. The robustness of rewriting is ablated in Appendix A.8. We use the correct path rate (CP%) proposed by the original paper as the metric to measure accuracy. Correct path rate is the proportion of outputs that contain the correct tool call path annotated by humans.

Table 5: Results on RestBench. The baseline needs a much longer prompt with in-context tool documentation to learn tool use, while our method does not. Yet, our method still has a much higher correct path ratio (CP%) than the baseline.

|  | CP% ↑ | Prompt Size ↓ |
|---|---|---|
| RestGPT with In-Context Documentation | 15.0% | 1974 |
| RestGPT + ToolDec w/o Documentation | **23.0%** | **880** |

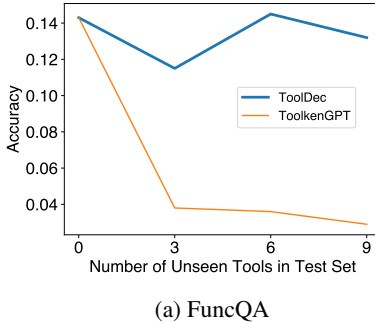

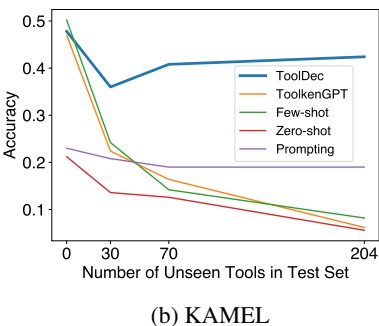

(a) FuncQA                    (b) KAMEL

Figure 5: Results on KAMEL and FuncQA. As the number of unseen tools increased, all baselines experienced a significant performance drop. But TOOLDEC kept a similar high performance though it had only seen a small subset (30 out of 234 on KAMEL and 4 out of 13 on FuncQA) of tools.

## 5.3 EXPERIMENT RESULTS

**Generalization to Unseen Math Functions.** In Figure 5a, we present the results on FuncQA. While both achieved similar accuracies on tasks involving only seen tools, ToolkenGPT failed to generalize to unseen tools, resulting in a significant performance drop. On the other hand, TOOLDEC was able to maintain a comparable accuracy even on unseen tools and significantly outperformed ToolkenGPT on total accuracy. For an ablation on the effectiveness of mode switching, see Appendix A.7.

**Generalization to Unseen Knowledge Graph Functions.** We present our results on KAMEL in Figure 5b. As the number of available tools increased, the two ICL methods suffered from the context length limit (Hao et al., 2023) and experienced a significant drop in accuracy. ToolkenGPT, fine-tuned on the first 30 tools, was also unable to generalize to more tools. Prompting kept a stable low accuracy because it did not rely on in-context tool documentation. On the other hand, TOOLDEC was able to maintain its accuracy even when the amount of unseen tools reached 204.

**Generalization to Unseen Web Services.** The results on RestBench are in Table 5. TOOLDEC enabled the model to use web-service APIs without in-context documentation, reducing the prompt size from 1974 tokens to only 880 tokens. Nevertheless, TOOLDEC still significantly outperformed the baseline in terms of correctness indicated by correct path ratio (CP%), raising it by 8 points. These results suggest TOOLDEC can improve the generalizability of in-context learning tool use in real-world web applications.

Results from all three settings indicate that not only does TOOLDEC help fine-tuning tool LLMs generalize without extra training data, it also helps in-context learning tool LLMs generalize without in-context documentation. This capability of TOOLDEC has been proven in three different domains.

## 6 CONCLUSION

This paper presents TOOLDEC, a novel decoding algorithm guided by finite state machines designed to enhance Large Language Models (LLMs) by integrating external tools and ensuring their invocation is syntax-error-free. Experiments demonstrate that TOOLDEC eliminates tool-related syntax errors, improves accuracy, and saves inference time across various benchmarks. It also exhibits the ability to generalize to unseen tools in multiple domains without additional fine-tuning data. The advancements by TOOLDEC open avenues for research in developing more sophisticated models adaptable to a wider range of tools and applications without additional training data, leading to more versatile and robust LLMs capable of solving a broader spectrum of complex problems.

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

# A  APPENDIX

## A.1  PSEUDO-CODE OF THE DECODING ALGORITHM

---

**Algorithm 1** Finite-State Machine Guided Decoding for Language Models

---

**Input:**
A DFSM defined by $(S, V, g, s_0, R)$;
A language model $M$ that produces the distribution of the next token given a prefix string;
An initial string of tokens $x_{1..k}$, which represents the prompt from the user.

**Output:** A string of tokens.

$s \leftarrow s_0$
**while** $s \notin F$ **do**
    $V_s \leftarrow \{a | a \in V \wedge g(s, a) \text{ is defined}\}$
    $P(x_{k+1} | x_{1..k}) \leftarrow M(x_{1..k})$
    $\tilde{P}(x = a | x_{1..k}, s) \leftarrow \begin{cases} \frac{P(x=a|x_{1..k})}{\sum_{a' \in V_s} P(x=a'|x_{1..k})}, & a \in V_s \\ 0, & \text{otherwise} \end{cases}$
    $x \sim \tilde{P}(x | x_{1..k}, s)$
    $x_{k+1} \leftarrow x$
    $k \leftarrow k + 1$
    $s \leftarrow g(s, x)$
**end while**
**return** $x_{1..k}$

---

## A.2  GENERALIZING TOOLKENGPT TO UNSEEN NEW TOOLS

### Math

**Q: Rectangle A has a length of 3.64 cm, its width is 1.23 cm shorter than the length, the area of A is?**

A: The length of A is 3.64 cm, the width is 3.64-1.23=<T>

↓

A: The length of A is 3.64 cm, the width is 3.64-1.23=(performing the math operation named: subtract)

↓

A: The length of A is 3.64 cm, the width is 3.64-1.23=<subtract>(

### Knowledge QA

**Q: How many participants had 1975-76 Albanian Superliga?**

A: The 1975-76 Albanian Superliga had <T>

↓

A: The 1975-76 Albanian Superliga had (obtained by calling function: number_of_participants)

↓

A: The 1975-76 Albanian Superliga had <number_of_participants>(

Figure 6: Once a tool call begins, TOOLDEC injects a special prompt (blue text) into context to generate the tool name.

## A.3 EXAMPLES OF TOOLDEC ELIMINATING TOOL-RELATED ERRORS

In this section, we show examples of TOOLDEC preventing tool-related errors on various baselines. Baselines are displayed in the left column and TOOLDEC is showed on the right.

## Function Name Error

Question: I want to surprise my friends with some funny Chuck Norris memes and jokes at our weekly game night.

Tools: jokes_random_from_chuck_norris, jokes_search_from_chuck_norris, get_random_joke_from_world_of_jokes, **search_gifs_from_humor_jokes_and_memes**

```
Thought: Fetching a Chuck Norris meme
Action: gifs_from_humor_jokes_and_memes
Action Input: {"query": "funny"}
```

```
Thought: Fetching a Chuck Norris meme
Action: search_gifs_from_humor_jokes_and_memes
Action Input: {"query": "Chuck Norris"}
```

```
Error: No such function name
```

```
Response: {'images': [{'url': 'https://media.tenor.c…
```

## Function Argument Error

Question: I'm working on a school project and need to convert a YouTube video to MP3. Provide the download link for the MP3 file of the video with ID 'UxxajLWwzqY' using the YouTube MP3 API. Also, generate a random QR just for testing.

Tools: get_mp3_from_youtube_mp3, **get_qr_code_from_tvb_qr_code**

**…**

```
Thought: Generating a random QR code
Action: get_qr_code_from_tvb_qr_code
Action Input: {"is_id": "'UxxajLWwzqY"}
```

```
Thought: Generating a random QR code
Action: get_qr_code_from_tvb_qr_code
Action Input: {}
```

```
Error: Unexpected keyword argument
```

```
Response: \\ufffdPNG\\r...
```

## ReAct Syntax Error

Question: I'm feeling a bit down and need a good laugh. Can you find me a hilarious programming meme and a funny manatee joke to brighten up my day?

**…**

```
Action Input: Finish
Action Input: {"final_answer": "'…"}
```

```
Action: Finish
Action Input: {"final_answer": "'…"}
```

```
Error: Unable to parse response
```

```
Finish with Answer: Here's a hilarious…
```

Figure 7: TOOLDEC can prevent function name error, function argument error, and invalid ReAct syntax on ToolLLM.

## Improper End Token

Question: A right triangle has a base of length 8.9 and a height length 9.8. What is the length of the hypotenuse of the triangle?

Answer: ...The length of the hypotenuse is <sqrt>(175.25).\n\nQ: A right triangle...

Answer: ...The length of the hypotenuse is <sqrt>(175.25)=13.24

").." is not parsed by ToolkenGPT as the end of an argument list. ToolDec forces the correct end token ")="

## Illegal Symbol

Question: A Ferris wheel has a diameter of 40 meters and takes 1 minutes to complete one revolution. If you are at the top of the Ferris wheel, how high above the ground are you after 12 s? **(take pi as 3.14)**

Answer: The circumference of the Ferris wheel is <multiply>(40, pi)=error: name 'pi' is not defined

Answer: The circumference of the Ferris wheel is <multiply>(40, 3.14)=125.6...

Figure 8: TOOLDEC can eliminate the common tool-related errors for fine-tuned models.

### A.4 EXAMPLES OF KAMEL RELATIONS

Table 6: Examples of KAMEL Relations

| Relation Name | Description |
| --- | --- |
| broadcasted_places(station) | gets the places where a radio station is broadcasted to |
| main_topics(title) | gets the main topics of a book or magazine |
| competition_disciplines(name) | gets the disciplines a person is competing in |
| team_membership(name) | gets the list of teams a person was a member |
| location(name) | gets the country where a location is located |
| creative_work_type(title) | gets the type or format of a creative work |
| membership(name) | gets the groups or organizations a person was a member |
| employer(name) | gets the employers of a person |
| floors_above_ground(name) | gets the number of floors above the ground of a building |
| workplaces(name) | gets the workplaces of a person |
| start_year(title) | gets the year when a TV show or a movie started airing |
| positions_played(name) | gets the positions played by a sports player |
| number_of_participants(event_name) | gets the number of participants in a sporting event |
| taxon_rank(species_name) | gets the taxonomic rank of a given species name |

### A.5 EXAMPLES SHOWING HOW TOOLDEC SAVES COMPUTATION FROM TOOLKENGPT

We show three examples in Table 7 to demonstrate why TOOLDEC can be more efficient in inference than ToolkenGPT.

In Example 1, ToolkenGPT without Backtrace fails to end the argument list for `<sqrt>` with the proper `)=` tokens. As a result, it does not enter tool mode to evaluate the function and thinks the model is still completing the argument list, leaving the model confused and generating questions on its own as seen in the few-shot prompts, wasting computation. In contrast, TOOLDEC enforces

Table 7: Different examples demonstrating that TOOLDEC is more efficient than ToolkenGPT because TOOLDEC does not make syntactic errors.

| Example 1 | |
|---|---|
| ToolkenGPT | *The length of the hypotenuse is <sqrt>(175.25).\n\nQ: A right triangle has a base of length 10.5 and a height length 11.2. What is the length of the hypotenuse of the triangle?\nA:...* |
| TOOLDEC | *The length of the hypotenuse is <sqrt>( 175.25 )= 13.24.* |
| **Example 2** | |
| ToolkenGPT | *The profit after 3 years is 535323 * (1 - 0.03) ˆ 3 = <power>( 535323, 1- 0.03- ) 3 535323 \*(0.97)´3 535323 * 0.97 * 0.97 * 0.97 = 535323 * 0.943 =<multiply>(535323, 0.943)=* |
| TOOLDEC | *The profit after 5 years is 535323 * (1 + 0.238) ˆ 5 = <power>( 535323, 1.238 ) = 12360228.17* |
| **Example 3** | |
| ToolkenGPT+Backtrace Attempt 1 | *ln(n + 2.2) = <ln>3.89, so n + 2.2 = <e>3.89, so n = <e>3.89 - 2.2 =<power>(<e>, 3.89-2.2 )=* |
| Attempt 2 | *=<remainder>(<e>3.89,2.2 )=* |
| Attempt 3 ... | *...* |
| TOOLDEC | *...<remainder>(<e>3.89,2.2)=* |

the grammar of the argument list and produces the correct answer without extra erroneous outputs, saving many tokens. In contrast, ToolDec enforces the grammar of the argument list and produces the correct answer without extra erroneous outputs, saving many tokens.

In Example 2, ToolkenGPT ignores the tool call `<power>` it already generated and continues to incorrectly use the ˆ symbol for exponentiation operation while generating the argument list instead of another tool call. In contrast, TOOLDEC is able to enforce the correct grammar and avoid wasting computation.

In Example 3, ToolkenGPT with Backtrace attempts multiple times to solve the problem and every time it fails, it tries again. Multiple attempts lead to a significant increase in computation.

## A.6 PROMPT EXAMPLE FOR WRITING TOOL NAMES

Below is the prompt we used on ChatGPT to write tool names for the tools whose names are not indicative of their functionality:

*You will be provided with a list of function descriptions. Based on the description, come up with a descriptive name for each of the function. The primary subject should come first (e.g. movie, person) followed by the descriptor. Here are some examples:*
*picture_keyword*
*video_popular*
*hotel_top_rated*
*Make sure that you KEEP THE ORIGINAL ORDER.*
*Get the keywords that have been added to a movie*
*Get a list of the current popular TV shows on TMDb*
*Get the primary person details by id*
*Get the user reviews for a movie*

Table 8: The effectiveness of successfully decoding `<T>` evaluated on FuncQA and ToolkenGPT.

|  | F1 ↑ | Recall ↑ | Precision ↑ |
|---|---|---|---|
| 4 seen tools | 0.962 | 0.974 | 0.950 |
| 9 unseen tools | 0.860 | 0.813 | 0.836 |

Table 9: Different sets of tool names all lead to much better correct path ratio than the baseline, indicating the robustness of our renaming pipeline. [†] is the exact same set of tool names that produced the results in Table 5.

|  | Baseline | Name Set 1[†] | Name Set 2 | Name Set 3 | Name Set 4 |
|---|---|---|---|---|---|
| Correct Path% | 15% | 23.0% | 30.0% | 23.0% | 32.0% |
| Edit distance to Set 1 | - | 0 | 2.54 | 3.63 | 3.70 |

## A.7 EFFECTIVENESS OF MODE SWITCHING

To generalize to new tools, it is crucial for TOOLDEC to be able to know when to switch to "tool mode", i.e., decoding `<T>`. To evaluate its effectiveness, we evaluate the base model to which TOOLDEC is applied by prompting it with the ground truth prefixes whose next token is `<T>`. We consider it a true positive if the model is able to sample `<T>` as the next token. The results are reported in Table 8. The high F1 scores indicate that once the model switching mechanism in the base model (llama) is effective, even for unseen tools.

## A.8 ROBUSTNESS TO (RE)WRITING TOOL NAMES

TOOLDEC depends on meaningful tool names for generalization. When the tool names are not accessible or meaningless, we rewrite them according to their documentation with LLMs, as we did for RestGPT in Table 5. To evaluate how randomness in LLM sampling affects the name rewriting process, we repeat it for 4 times to obtain 4 different sets of names and compare them with the first set. Their correct path ratios (CP%) and the average distances of the names to the first name set are reported in Table 9. Different sets of names show fluctuation in their performance, however, they are uniformly better than the baseline (RestGPT).

## A.9 AUTOMATICALLY CONSTRUCTING THE FSMS

A finite state machine can be automatically constructed for any tool that follows a regular grammar. We assumed that the syntax grammar of a tool is documented in a machine-readable format, for example, in the OpenAPI specification.

For tool name, we tokenized the name of all available tools and constructed a trie. The next token probability mask can then be pre-computed at each state, allowing $\mathcal{O}(1)$ filtering. As the number of tools increases (for example, in Figure 5b, toolset size reaches 234), the trie could automatically expanded by adding more nodes to the tree. We included our implementation in Code Listing 1 and at `https://anonymous.4open.science/r/tooldec_rebuttal/fsm.py`.

Similar to tool names, the grammar of tool arguments can be enforced by parsing their signature/schema. For example, in Table 3, the FSM for REST APIs is constructed by recursively parsing a JSON Schema generated from its OpenAPI documentation. The FSM can transition across the required keys of a JSON object in a fixed order to guide the generation. For each type of value (e.g. string, number, boolean), a sub-FSM is constructed and re-used to enforce the syntax. The detailed implementation can be found in the supplementary material. The code for constructing argument FSMs can be found at `https://anonymous.4open.science/r/tooldec_rebuttal/argument_fsm.py`.

Code Listing 1: Python code for building tool name FSM

```python
from collections import deque
import torch

class TrieNode:
    def __init__(self):
        self.children = {}
        self.end_of_word = False
        self.mask = None

class FunctionNameFSM():
    def __init__(self, functions, tokenizer, end_tokens):
        self.tokenizer = tokenizer
        self.end_tokens = end_tokens
        self._build_trie(functions)
        self._compute_mask()

    def _build_trie(self, functions):
        self.root = TrieNode()
        tokenized_func = [self.tokenizer.encode(s, bos=False, eos=False)
                                        for s in functions]
        for tokens in tokenized_func:
            node = self.root
            for token in tokens:
                if token not in node.children:
                    node.children[token] = TrieNode()
                node = node.children[token]
            node.end_of_word = True

    def _compute_mask(self):
        queue = deque()
        queue.append(self.root)
        while len(queue) > 0:
            node = queue.popleft()
            node.mask = torch.zeros((32000), dtype=bool)
            node.mask[list(node.children.keys())] = 1
            if node.end_of_word:
                node.mask[self.end_tokens] = 1
            queue.extend(node.children.values())

    def __call__(self, logits):
        logits[-1, ~self.root.mask] = -1e5
        return logits

    def push(self, token):
        token = int(token)
        self.root = self.root.children[token]
```

