# OpenReview forum: "ToolDec: Syntax Error-Free and Generalizable Tool Use for LLMs via Finite-State Decoding"
_ICLR.cc/2024/Conference — Submitted to ICLR 2024_

### Official Review · Reviewer_uQYf · 2023-10-31

**Soundness:** 2 fair
**Presentation:** 2 fair
**Contribution:** 1 poor
**Rating:** 5
**Confidence:** 4

**Summary:**

This paper proposes to use Finite State Machine (FSM) for LLM decoding to constrain the search space and reduce the syntax error for the tool use by LLM. The idea is very simple: It is basically to construct an FSM from the tool signature and introduce a special symbol to switch the normal text mode and the tool use mode and use the FSM to constrain the search space of LLM. As long as the text to use tools is generated in the tool use mode and we can assume that all paths in the mode are all valid, it is guaranteed that there is no syntax error. The experimental results confirm that the proposed method can make the number of syntax error generated by LLM be zero.

**Strengths:**

This paper shows that LLMs still require an external knowledge to constrain the search space for tool use and existing methods such as finetuning and in-context learning are not enough. It shows that the type of errors (syntax errors) can be addressed by an adoption of a simple FSM. It is shown that it is true for the settings the method was tested for.

**Weaknesses:**

1. Novelty

It is essentially about constraining the search space of a language model by a grammar. It is definitely expected that the use of a grammar can reduce syntax errors if we know that the output needs to follow the grammar. I feel that it is a known technique but not a novel finding although probably it has not been applied for LLMs yet. It does not necessarily need to be theoretical, but I would probably at least want to see deeper discussions on why LLM has limitations without such external knowledge. Does a much stronger LLM have the same problem?

2. Complexity

A good thing about LLMs is that the input and output are both plain text and the mechanism is very simple. This technique is against the simplicity. It says the FSM can be automatically constructed, but I am not sure if it is always the case for more complex tools. Defining FSM manually could be tedious and error-prone for complex ones. Decoding with an external FSM will add additional complexity to the system although this could be standardized by for example open tools.

**Questions:**

I would love to see the list of tools and their signatures used in the experiments. It should help to understand the complexity of the problem addressed by this paper. I would love to see the prompt used for both baselines and the proposed method. I should be able to find some of the information by looking into previous studies or the original datasets, but I think having it in the paper should be valuable. It could be in appendix.

**Details Of Ethics Concerns:**

No concern.

---

> ### Author Response · Authors · 2023-11-17
> **Response to Reviewer uQYf 1**
>
> Thank you for the comments! We would like to address the issues you raised here.
>
> ### 1. Novelty
>
> > a known technique but not a novel finding
>
> Grammar-constrained decoding is indeed a known technique for reducing syntax errors, we will discuss more about it in the revision. However, ToolDec goes beyond elimination of syntax errors.
>
> First, ToolDec not only constrains the generation of tool calls, it also guides the reasoning of LLMs by switching between an unconstrained "text mode" and a constrained "tool mode". This is different from previous grammar-constrained decoding algorithms that work mostly for structured outputs.
>
> Second, the novelty of our findings also lies in better generalization, as we have discussed in Section 5. Without in-context documentation of tools or further fine-tuning, ToolDec can enable LLMs to generalize to tools that they have not seen in the training data and know when and how to use them. As Figure 5(b) shows, ToolDec can perform well even when there are as many as 204 unseen tools in the test set.
>
> ### 2. Why much stronger LLMs can still have limitations
>
> > why LLM has limitations without such external knowledge. Does a much stronger LLM have the same problem?
>
> Yes, even a much stronger LLM can have the same problem.
>
> To clarify, there are two problems that ToolDec solves: correct use of seen tools and generalization to unseen tools. While the former might be solved with a much stronger LLM, the latter is inherent to LLMs no matter how strong. Because there will always be changes to the existing tools and emergence of new tools after LLMs are trained. Without seeing them in training data, even the most powerful LLMs cannot know when and how to use new, unseen tools. Putting the documentation of new tools in the prompt or further fine-tuning may solve the problem, but not as efficiently as ToolDec.
>
> Our Table 3 provides evidence for this argument. One would generally agree that ChatGPT is a stronger LLM than Llama (from which ToolLLM was fine-tuned). Yet ChatGPT still had many syntax errors (error rate=8%) in tool use and was outperformed by ToolDec+ToolLLM (error rate=0). Note that the ChatGPT baseline in Table 3 was given the documentation of the tools. Without in-context documentation, it could only be worse.

---

> ### Author Response · Authors · 2023-11-17
> **Response to Reviewer uQYf 2**
>
> ### 3. Automatic construction of FSM can work for more complex tools
>
> > It says the FSM can be automatically constructed, but I am not sure if it is always the case for more complex tools.
>
> Yes, the FSM can always be automatically constructed.
>
> Our construction pipeline works for anys tool as long as their specifications follow to some standards (like [OpenAPI](https://www.openapis.org/)). These standards are very expressive and can specify complex tools. Therefore, we can automatically construct FSMs for complex tools.
>
> We can construct FSMs for any complex tool names. As discussed in Section 3.2, the data structure, trie, can serve as an abstract FSM responsible for enforcing tool name grammar. As the number of tools increases (in Section 5.1 with KAMEL, toolset size reaches 234. See [here](https://anonymous.4open.science/r/tooldec_rebuttal/prompts/kamel.txt) for a full list of tool names), the trie could scale automatically by adding more nodes to the tree.
>
> We can construct FSMs for any tool specified by the [OpenAPI specification](https://www.openapis.org/). In our ToolLLM and RestGPT experiments, we tested ToolDec with REST APIs, which follow OpenAPI specifications. So we can construct an FSM by by recursively parsing the APIs' argument schema, even for a tool whose specification is as complex as this [example](https://anonymous.4open.science/r/tooldec_rebuttal/tools/toolllm_example.json). OpenAPI specifications are used by most service providers and online API Hubs to describe their APIs, which cover many complex tools.
>
> ### 4. The additional complexity is minimal
>
> > Decoding with an external FSM will add additional complexity to the system although this could be standardized by for example open tools.
>
> We argue that the additional complexity is minimal and worthy.
>
> In terms of computation complexity, the only extra GPU operation created by ToolDec is masking the invalid tokens at each step. This is neglectable compared to the large amount of computation needed in attention blocks (large matrix multiplication, softmax, etc). In practice, ToolDec actually runs much faster than the baselines because it doesn't make syntactic mistakes.
>
> In terms of the complexity added to the system, ToolDec is model-agnostic and can be plugged in place of the original decoding algorithm without interfering with other components, as shown in our successful integration with multile baselines.
>
> > I would love to see the list of tools and their signatures used in the experiments.
>
> Please use [this link](https://anonymous.4open.science/r/tooldec_rebuttal/tools) to access the tools evaluated in the experiments. We were unable to attach the full toolset of ToolLLM due to its enourmous size (10K+), but we included some examples. Prompts are attached [here](https://anonymous.4open.science/r/tooldec_rebuttal/prompts). The prompts we used are directly taken from our baselines.
> We will provide these details in the revision.
>
> If you have any other questions or comments, we will be happy to further clarify.

---

> > ### Author Response · Authors · 2023-11-20
> > **Follow-Up: Seeking Further Feedback**
> >
> > Dear Reviewer,
> >
> > We wanted to express our gratitude for the time and effort you have put into reviewing. Your comments and suggestions have been truly valuable in helping us refine our work.
> >
> > We would like to know if we've addressed all your concerns. Please let us know if any further clarification or additional information is needed from our end.
> >
> > Best,
> >
> > Authors

---

> ### Author Response · Authors · 2023-11-22
> **Follow-Up: Seeking Feedback Before Discussion Closes**
>
> Dear Reviewer uQYf,
>
> Since the discussion period is closing today, we would like to know if our resposne was able to address your questions and would like to take the last chance in providing any clarification.
>
> Additionally, we would like to point out again that beside the strengths you mentioned, ToolDec's generalization ability should not be overlooked. By reducing the search space, it enables LLMs to select tools entirely based on the semantics of tool name. This enables the usage of a large tool set (up to 204 in Figure 5(b)) and complex tools (Table 5) without fine-tuning or in-context learning and achieved superior performance to the baseline.
>
> We have revised our paper to make this more clear. Meanwhile, we hope that you can take into consideration when evaluating ToolDec. Thank you again for your time and effort in helping us improving our research.
>
> Looking forward to your response,
>
> Authors

---

> ### Comment · Reviewer_uQYf · 2023-11-23
> **Thank you for the rebuttal**
>
> Thank you for the detailed response. I'm going to definitely reflect the rebuttal to make my final decision. Thanks again for your hard work.

---

### Official Review · Reviewer_R8ud · 2023-10-31

**Soundness:** 3 good
**Presentation:** 3 good
**Contribution:** 2 fair
**Rating:** 5
**Confidence:** 5

**Summary:**

It is desirable in some applications to augment instruction-tuned language models with the ability to call tools, such as calculators, in responding to user prompts. This can mitigate certain inherent limitations of the language model as well as augment it with novel capabilities. However, "teaching" a language model to use tools is difficult, due to the lack of suitable training data. This paper proposes an inference-only constrained decoding approach using finite-state machines (FSM). By hand-crafting a FST for each tool, it is possible to eliminate syntax errors that can occur when relying solely on in-context learning. The approach is applied “on top” of two existing tool-augmented language models, ToolLLM and ToolkenGPT, showing improvements in the ability to correctly apply tools. In a further experiment, generalization to novel tools is evaluated, showing that the approach can successfully be adapted to new tools.

**Strengths:**

* Experimental validation that enforcing prefix-checkable constraints on generation can result in more effective tool use for a relatively large set of tools.
* The decoding approach is validated for several different LLM (ToolkenLLM, RestGPT, and ToolLLM), and appears to improve the in-context learning ability of the LLM (S4.3).
* Approach maintains levels of accuracy even with increasing numbers of unseen tools in the test set (Figure 5).

**Weaknesses:**

* Unclear why general machinery of FSM is necessary when the approach amounts to constrained decoding using prefix-checkable constraints on next token generation. The paper states “Note that in practice, it’s not necessary to explicitly construct this FSM. Any grammar checker that tells the set of valid next tokens suffice.” Perhaps there could be better motivation for using FSM?
* In some ways, the approach seems like a step backwards to expert-based AI, in that the improvements from the proposed approach appear largely to be the result of hand-crafting decoding constraints.
* Related to the above concern, it’s unclear how the proposed approach was validated. Was the hand-crafted decoding approach tailored to perform on test data?

**Questions:**

* Table 4 contains timing results in seconds. Were all methods equally optimized?
* The "fine-tuning" terminology is confusing since the approach consists of hand-crafted decoding constraints; there's no parameter fine-tuning involved in ToolDec as I understand. So does this refer to the LLM + ToolDec only being evaluated on unseen tools?

---

> ### Author Response · Authors · 2023-11-17
> **Response to Reviewer R8ud**
>
> We thank the reviewer for their comments.
>
> ### 1. Motivation for using FSM
>
> > Perhaps there could be better motivation for using FSM?
>
> We choose Finite-State Machines (FSMs) because they are simple and efficient yet expressive to cover most scenarios for function calls.
>
> We will make the distinction between the abstract model of computation and the actual implementation clearer in the revision." It would be possible to extend to more expressive automata in the future.
>
> ### 2. Our constraints are *not* hand-crafted
>
> > a step backwards to expert-based AI ... improvements from the proposed approach appear largely to be the result of hand-crafting decoding constraints.
>
> Sorry for causing the misunderstanding. The decoding constraints are not hand-crafted. Instead, they are constructed automatically from the signatures of the tool functions. Our automatic pipeline parses the signatures to obtain constraints for function names and function arguments. This is similar to the program analysis techniques [1] used by IDEs to give code completion suggestions such as variable names.
>
> We use a trie structrue [2] to construct the FSM for tool names (Sec. 3.2). The algorithms for building tries, inserting strings, and searching are easy to implement. We list our code for automatically building the tool name FSM with trie [here](https://anonymous.4open.science/r/tooldec_rebuttal/fsm.py). As the number of tools increases (in Section 5.1 with KAMEL, toolset size reaches 234. See [here](https://anonymous.4open.science/r/tooldec_rebuttal/tools/kamel_relations.json) for a full list of tool names), the trie could automatically expanded by adding more nodes to the tree.
>
> Similar to tool names, the grammar of tool arguments can be enforced by parsing its signature/schma. For example, the FSM for REST APIs can be constructed by recursively parsing a JSON Schema generated from its OpenAPI documentation. Please see [here](https://anonymous.4open.science/r/tooldec_rebuttal/tools/toolllm_example.json) for an example.
>
> #### Reference
>
> [1] Nielson, Flemming, Hanne R. Nielson, and Chris Hankin. Principles of program analysis. Springer, 2015.
> [2] Fredkin, Edward. "Trie memory." Communications of the ACM 3.9 (1960): 490-499.
>
>
> ### 3. Constraints are not tailored to perform on test data
>
> > Was the hand-crafted decoding approach tailored to perform on test data?
>
> No, the decoding constraints were not hand-crafted (as pointed out in response 2), and they were not tailored to test data.
>
> During evaluation, our system has access to the whole tool inventory, but not to the specific tool used by each test example. We constructed one single set of constraints for all the tools involved without any test set-specific or case-specific hand-crafting. The decoding constraints were constructed entirely from tool documentations and no assumptions about their actual usecases in the testing dataset were made. Therefore, every test case in a test set has the same decoding constraints.
>
>
> ### 4. All methods are equally optimized in Table 4
>
> > Were all methods equally optimized?
>
> Yes. We utilized the same model checkpoint for ToolDec and the two timed baselines (ToolkenGPT, ToolkenGPT+Backtrace). All original code from the baselines was retained as is, with the only difference being that ToolDec was added on top of the original decoding algorithm. We assessed the methods using the exact same hardware and software environment.
>
> Compared to the ToolkenGPT without backtrace, ToolDec requires less time as it prevents the generation of a long and invalid chain of tokens in place of tool arguments. The decrease in inference time is even more pronounced compared to ToolkenGPT + Backtrace because backtrace can cause ToolkenGPT to repeatedly try different tools with invalid arguments.
>
> We will clarify this issue in the revised version.
>
> ### 5. "fine-tuning" terminology
>
> >  the approach consists of hand-crafted decoding constraints; there's no parameter fine-tuning involved in ToolDec
>
> First, as mentioned before, there is no "hand-crafted" decoding constraint in ToolDec.
>
> Second, "fine-tuning" is not involved in ToolDec. One of our baselines, ToolkenGPT uses fine-tuning. Its base model (llama) is fine-tuned on a limited set of tools and then evaluated on unseen tools. Similarly, the exactly same base model (llama) for ToolDec was also fine-tuned on this small set to learn when to use a tool, to ensure fair comparison.
>
> Thank you again for your time! We are happy to further clarify should you have any other questions.

---

> > ### Author Response · Authors · 2023-11-20
> > **Follow-Up: Seeking Further Feedback**
> >
> > Dear Reviewer,
> >
> > We wanted to express our gratitude for the time and effort you have put into reviewing. Your comments and suggestions have been truly valuable in helping us refine our work.
> >
> > We would like to know if we've addressed all your concerns. Please let us know if any further clarification or additional information is needed from our end.
> >
> > Best,
> >
> > Authors

---

> > > ### Author Response · Authors · 2023-11-23
> > > **Follow-Up: Seeking Feedback Before Discussion Closes**
> > >
> > > Dear Reviewer R8ud,
> > >
> > > Since the discussion period is closing today, we would like to know if our response was able to address your questions and would like to take the last chance to provide any clarification.
> > >
> > > We would like to emphasize again that our FSM was not hand-crafted and kindly ask you to take this into account when evaluating our paper.
> > >
> > > We have revised our paper to make this more clear. Meanwhile, we hope that you can take this into consideration when evaluating ToolDec. Thank you again for your time and effort in helping us improve our research.
> > >
> > > Looking forward to your response,
> > >
> > > Authors

---

> > > > ### Comment · Reviewer_R8ud · 2023-12-03
> > > > **Thanks!**
> > > >
> > > > Thank you for the detailed response which has addressed some of my concerns. I have increased my recommendation accordingly.

---

### Official Review · Reviewer_ydwb · 2023-10-31

**Soundness:** 3 good
**Presentation:** 3 good
**Contribution:** 3 good
**Rating:** 6
**Confidence:** 4

**Summary:**

The paper introduces a decoding method for LLMs to use external tools while avoiding syntax errors. The core idea is to constrain the models in only able to decode from a selected set of valid tokens that conform with the tool signatures. The method is compatible with existing LLM tool-use schemes (in-context learning and finetuning), and empirically removes syntax errors and thus achieving improved performances.

**Strengths:**

- The FSM guided decoding method is intuitive and suitable for solving the syntax errors.
- The proposed method is compatible with existing LLM tool-use schemes, i.e., both finetuning or in-context learning.
- The method has shown to be empirically effective in eliminating syntax error, and leads to performance improvements.

**Weaknesses:**

- The FSM construction may require careful curation. For example, how does one decide what's the best naming for a tool? Are LLMs robust to the name changes? Also, what would the process be like for one to construct the FSMs for a large collection of tools? Would it be done through parsing the tool documentations? It'd be helpful if the authors provide more discussion here.
- It is not clear to me as to how ToolDec can enable generalization to new tools? While adding new FSM (for the new tool) can ensure the LLM uses the new tool in a syntactically correct way, the FSM itself does not provide sufficient information on when the tool should be invoked. Current generalization then seems to only depend on LLM's language prior, and thus related to above, it's tool use performance can largely depend on the proper naming of the tools.
- Following from above, it'd be interesting to see an experiment testing the robustness of ToolDec by assigning tool names that not are not semantically meaningful.

**Questions:**

- Why would ToolDec be faster at inference compared to ToolkenGPT? Could the authors provide more explanation?

---

> ### Author Response · Authors · 2023-11-17
> **Response to Reviewer ydwb 1**
>
> Thank you for the comments! We would like to address the issues you raised here.
>
> ### 1. FSM construction
>
> > what would the process be like for one to construct the FSMs for a large collection of tools? Would it be done through parsing the tool documentations?
>
> Yes. We would like to emphasize that this process is done automatically by parsing the tool documentations. For example, as discussed in Section 3.2, we use a trie structrue to construct the FSM for tool names. The algorithms for building tries, inserting strings, and searching are easy to implement. We list our code for automatically building the tool name FSM with trie [here](https://anonymous.4open.science/r/tooldec_rebuttal/fsm.py). As the number of tools increases (in Section 5.1 with KAMEL, toolset size reaches 234. See [here](https://anonymous.4open.science/r/tooldec_rebuttal/tools/kamel_relations.json) for a full list of tool names), the trie could automatically expanded by adding more nodes to the tree.
>
> Similar to tool names, the grammar of tool arguments can be enforced by parsing its signature/schma. For example, the FSM for REST APIs can be constructed by recursively parsing a JSON Schema generated from its OpenAPI documentation. Please see [here](https://anonymous.4open.science/r/tooldec_rebuttal/tools/toolllm_example.json) for an example.
>
>
> ### 2. How ToolDec enables generalization to new tools
>
> > Current generalization ... can largely depend on the proper naming of the tools.
>
> Current generalization ability of ToolDec is indeed based on the assumption that the tools are named properly. In fact, one of the key observations of our research is that proper naming on its own can eliminate the need for training data and in context documentations, as shown in Figure 5(b).
>
> The assumption that tools have semantically meaningful names is reasonable. Naming conventions for identifiers have long been established for different programming languages in both industry [1] and open-source communities [2]. Therefore, for most popular libraries, we can assume proper naming.
>
> > how does one decide what's the best naming for a tool?
>
> Following the previous point, we argue that even in the case where functions are not named or poorly named, LLMs can be utilized to create semantically meaningful function names, as evident in our RestGPT experiment (Table 5). In this experiment, we prompted ChatGPT to create meaningful names for the REST APIs. The prompt used to generate tool names is [here](https://anonymous.4open.science/r/tooldec_rebuttal/generate_tool_name.txt). We found that the two most important criteria are having the primary object (e.g. movie, tv_show, etc) at the beginning of the name and using only singular nouns.  This renaming ability has also been explored by other existing work [3].
>
> > ... Are LLMs robust to the name changes?
>
> To evaluate the robustness of LLMs with respect to name changes, we use the exact same prompt mentioned in the last response and sample the names of tools multiple times.
>
> Each time, we evaluate the performance when the resampled tool names are used and compute the average edit distance between the newly sampled names and the names in the first sample. The first set of sampled names are the ones that produced the results in Table 5.
>
> |                   | Sample 1 (Directly taken from Table 5, row 2)  | Sample 2 | Sample 3 | Sample 4 |
> |-------------------|-------|----------|---------|----------|
> | Accuracy          | 23.0% | 30.0%    | 23.0%   | 32.0%    |
> | Avg Edit Distance | 0   | 2.54     | 3.63    | 3.70     |
>
> As shown by the table, the accuracy of our method does flucuat with resampled names. However, even the worst name samples can still lead to much better performance than the RestGPT baseline.

---

> ### Author Response · Authors · 2023-11-17
> **Response to Reviewer ydwb 2**
>
> ### 3. Why ToolDec is faster than ToolkenGPT
>
> > Why would ToolDec be faster at inference compared to ToolkenGPT? Could the authors provide more explanation?
>
> As reported in Table 4, **ToolkenGPT+ToolDec** is faster than **ToolkenGPT w/o Backtrace** (row 4, 7.76s) and **ToolkenGPT w/ Backtrace** (row 5, 10.39s).
>
> Compared to the baselines, ToolDec uses less time because it prevents the generation of a long and invalid chain of tokens in place of tool arguments. We demonstrate this with 3 examples.
>
> #### Example 1
>
> In the following example, **ToolkenGPT w/o Backtrace** fails to end the argument list for `<sqrt>` with the proper `)=` token. As a result, it does not enter tool mode to evaluate the function and thinks the model is still completing the argument list, leaving the model confused and generating questions on its own as seen in the few-shot prompts, wasting computation.
>
> > ...The length of the hypotenuse is **<sqrt>(** 175.25).\\n\\nQ: A right triangle has a base of length 10.5 and a height length 11.2. What is the length of the hypotenuse of the triangle?\\nA:...
>
> In contrast, ToolDec enforces the grammar of the argument list and produces the correct answer without extra erroneous outputs, saving many tokens.
>
> > ...The length of the hypotenuse is **<sqrt>(** 175.25 **)=** 13.24.
>
> #### Example 2
>
> Here we have another example, where ToolkenGPT ignores the tool call `<power>` it already generated and continues to incorrectly use the `^` symbol for exponentiation operation while generating the argument list instead of another tool call.
>
> > The profit after 3 years is 535323\*(1-0.03)\^3=**<power>(** 535323, 1-0.03)\^3=535323\*(0.97)\^3=535323\*0.97\*0.97\*0.97=535323\*0.943=<multiply>(535323, 0.943 **)=**
>
> In contrast, ToolDec is able to enforce the correct grammar and avoid wasting computation.
>
> > The profit after 5 years is 535323*(1+0.238)^5=**<power>(** 535323,1.238 **)=** 12360228.17
>
> #### Example 3
>
> The decrease in inference time is even more significant compared to **ToolkenGPT w/ Backtrace** because backtrace can lead to ToolkenGPT repeately trying different tools with invalid arguments.
>
> In the following example, **ToolkenGPT w/ Backtrace** first tries to use the `<power>` tool and fails.
>
> > ln(n+2.2)=<ln>3.89, so n+2.2=<e>3.89, so n=<e>3.89-2.2=**<power>(** <e>, 3.89-2.2 **)=**
>
> Then it retries the `<remainder>` tool.
>
> >...**<remainder>(** <e>3.89,2.2 **)=**
>
> Then it retries another tool until it succeeds in generating one correct tool call.
>
> If you have any other questions or comments, we will be happy to further clarify.
>
> ### Reference
>
> [1] Google. "Google C++ style guide." url: https://google.github.io/styleguide/cppguide.html.
>
> [2] Torvalds, Linus. "Linux kernel coding style." Also available as https://www.kernel.org/doc/Documentation/CodingStyle (2001).
>
> [3] Anonymous. (2023). Improving code style for accurate code generation. In Submitted to The Twelfth International Conference on Learning Representations. https://openreview.net/forum?id=maRYffiUpI

---

> > ### Author Response · Authors · 2023-11-20
> > **Follow-Up: Seeking Further Feedback**
> >
> > Dear Reviewer,
> >
> > We wanted to express our gratitude for the time and effort you have put into reviewing. Your comments and suggestions have been truly valuable in helping us refine our work.
> >
> > We would like to know if we've addressed all your concerns. Please let us know if any further clarification or additional information is needed from our end.
> >
> > Best,
> >
> > Authors

---

> > > ### Comment · Reviewer_ydwb · 2023-11-20
> > > **Thank you to the authors for the response**
> > >
> > > Thank you for the response. It answered my questions. I am however not very convinced by relying generalization solely on tool names. I would remain my original score.

---

> > > > ### Author Response · Authors · 2023-11-21
> > > >
> > > > We would like to further clarify how ToolDec improves generalization.
> > > >
> > > > First, ToolDec made "tool selection depending on tool names and signatures" possible for the first time, an approach that is not only effective but also efficient as demonstrated by the superior results on RestGPT (Table 5). It's worth noticing that the baseline we are comparing to had access to all tool documentations in the context, but still performed worse than ToolDec. This indicates that in some scenarios depending on tool names for selection is indeed more reliable than depending on documentation.
> > > >
> > > > Second, ToolDec does not have to depend solely on tool names, and it is not mutually exclusive with existing generalization methods.
> > > > In some scenarios where depending on tool names may be not enough, it is fine to combine ToolDec with existing method such as in-context documentation, thus depending on both tool names and in-context docs.
> > > > Our ToolLLM experiments indicate that this combination can lead to even better generalization.
> > > > In the following Table (Table 3 in the  paper), ToolDec improved the effectiveness of in-context documentation on I2-Category and I3-Instruction, two test sets containing tools **unseen** during training. This underscores ToolDec's wide applicability because adding it on top of existing generalization methods can still improve the performance.
> > > >
> > > > |                         | I2-Category    |        |              | I3-Instruction |        |              |
> > > > |-------------------------|----------------|--------|--------------|----------------|--------|--------------|
> > > > |                         | Pass ↑         | Win ↑  | Tool Error ↓ | Pass ↑         | Win ↑  | Tool Error ↓ |
> > > > | ChatGPT                 | 39.0%          | -      | 8.0%          | 29.0%          | -      | 7.0%          |
> > > > | ToolLLM                 | 32.0%          | 44.0%  | 39.0%         | 18.0%          | 44.0%  | 47.0%         |
> > > > | ToolLLM + Fuzzy Matching| 38.0%          | 48.0%  | 19.0%         | 23.0%          | 49.0%  | 23.0%         |
> > > > | ToolLLM + ToolDec | **42.0%** | **55.0%** | **0.0%** | **34.0%** | **60.0%** | **0.0%** |
> > > >
> > > > Thank you again for your timely response! We would really love to hear more from you about how specifically we can make our case stronger and more convincing.

---

### Official Review · Reviewer_7DQS · 2023-11-01

**Soundness:** 4 excellent
**Presentation:** 4 excellent
**Contribution:** 3 good
**Rating:** 8
**Confidence:** 4

**Summary:**

The paper introduces TOOLDEC, a novel approach for improving the performance of large language models (LLMs) when using external tools. The method tries to avoid generating syntactically invalid tool calls in these approaches. TOOLDEC is a finite-state machine-guided decoding algorithm, which designs to work with any tool-augmented LLM and ensures the generation of valid tool names and type-conforming arguments. Notably, TOOLDEC empowers LLMs to select tools solely based on their names, eliminating the need for fine-tuning or in-context documentation.

**Strengths:**

Strengths:
This paper proposes the finite-state machine-guided decoding algorithm, which reduces the errors during calling tools. It is a clear and simple method to restrict the decoding space.
The experimental results show that the method is effective in the tool learning task, which significantly reduces name errors.

**Weaknesses:**

Weaknesses:
Even though the model achieves significant improvements, it is unclear the language and tool mechanism switching. I think the switching effectiveness should be evaluated and whether the <T> token can be appropriately decoded.
The augment errors are zero. However, this reason may lie in that the existing tool learning benchmark is a little easy. If the input is a more complex problem and contains several numbers, the argument can also be wrong. The zero error rate should be carefully claimed.

**Questions:**

N/A

---

> ### Author Response · Authors · 2023-11-17
> **Response to Reviewer 7DQS**
>
> We thank the reviewer for their comments.
>
> ### 1. Switching between language and tool
>
> > it is unclear the language and tool mechanism switching
>
> We would like to clarify the switching mechanism here. The switching mechanism we use can be adapted to different planning strategies.
>
> For ToolLLM, the baseline used ReAct to plan the reasoning process. Specifically, they explicitly prompt the model with `Thought:` `Action:` `Action Input:`. `Thought:` switches the model to language mode, while `Action:` and `Action Input:` asks for tool name and tool arguments. We follow the baseline to use these tokens as `<T>` to switch betwen modes.
>
> For ToolkenGPT, the baseline has a separate token for each single tool. When the token for a particular tool is sampled, the model switches to the tool mode. Similarly, we use one single token `<T>` for all tools.
>
> > the switching effectiveness should be evaluated and whether the <T> token can be appropriately decoded.
>
> Thank you for the suggestion.
>
> Here we report the F1 accuracy of decoding <T> evaluated on the test set. At each decoding step, if the ground truth and model output are both <T>, it's considered a true positive. If the ground truth is <T> and the model output is not, it's considered a false negative.
>
> Test set with tasks using the same 4 tools used in training:
>
> | F1    | Recall | Precision |
> |-------|--------|-----------|
> | 0.962 | 0.974  | 0.95      |
>
> Test Set with tasks using 9 unseen tools:
>
> | F1    | Recall | Precision |
> |-------|--------|-----------|
> | 0.860 | 0.813  | 0.836     |
>
> The high F1 scores indicate that our mechanism for switching between tool mode and text mode can effectively do so.
>
> Besides the F1 accuracy, we also argue that only accurately decoded <T> can lead to correct tool selection and the significant improvement in the downstream performance reported in our experiments.
>
> It is also worth noticing that ToolDec is compatible with other planning frameworks (when to use tool), such as the coarse-to-fine framework used in our RestGPT framework.
>
> ### 2. The argument error rate is still zero for more complex tools
>
> > If the input is a more complex problem and contains several numbers, the argument can also be wrong. The zero error rate should be carefully claimed.
>
> First, we would like to clarify that the zero argument error rate is theoretically guaranteed. We define an argument error as the model generating arguments that do not follow the syntax of argument types. For example, if there's a tool `int add(int a, int b)`, as long as the model is generating two integers as the arguments, there is no argument error. As we discussed in Section 3, the FSM we constructed is an exact description of the tool syntax, hence our claim about zero error rate is guaranteed.
>
> Second, we would like to point out that our evaluation experiments already involve complex tools with many arguments and ToolDec was able to achieve zero argument error for these tools. For example, our Figure 4 compares ToolDec with ToolLLM on the ToolBench dataset. [Here](https://anonymous.4open.science/r/tooldec_rebuttal/tools/toolllm_example.json) are some complex tools in the dataset. One particular API for "Quote for postcode in OCA e-Pack" has 7 required arguments. 4 of which are numbers.
>
> Thank you again for your time! We are happy to further clarify should you have any other questions.

---

> > ### Comment · Reviewer_7DQS · 2023-11-20
> >
> > I do not have further problem. This work is more like a compiler for tool learning. It is necessary. The last question I mean that such a compiler is suitable for some usage scenarios. If we want to feed a string parameter (not appear in existing datasets but I think it also should be considered), the method will be degenerated due to the mechanism of ToolDev is different with a vanilla compiler. Thus, I suggest that the authors can claim more specifically.

---

> > > ### Author Response · Authors · 2023-11-20
> > >
> > > Thank you for the response!
> > >
> > > We agree that string parameters are important for tools, and we should consider them in our method.
> > >
> > > However, we argue FSMs are able to describe the grammar of string literals like `"Hello World"`. The syntax of string literals can be described using regular expressions like the one [here](https://stackoverflow.com/questions/46290618/regex-for-strings-in-c). Regular expressions and finite automata are known to be equivalent [1].
> > >
> > > We will discuss how we can process string parameters in the revision.
> > >
> > > [1] Kleene, S. C. "REPRESENTATION OF EVENTS IN NERVE NETS AND FINITE AUTOMATA¹." Automata Studies: Annals of Mathematics Studies. Number 34 34 (1956): 3.

---

### Author Response · Authors · 2023-11-22
**General Comments on Our Revision**

We thank all reviewers for their thoughtful feedbacks; they have been very helpful in improving the quality of our research.

We have just uploaded a revised version of our paper, in which the changed parts are highlighted.

The changes in our revision can be summarized as the following:

1. Since the construction of our FSMs is mentioned by multiple reviewers (ydwb, R8ud, and uQYf), we included a section in **Appendix A.9** to explain how FSMs are constructed automatically from the tool documentation. Note that this process is applicable to any tools that follow certain standards, like OpenAPI specifications. We also release the pipeline construction code at [https://anonymous.4open.science/r/tooldec_rebuttal/](https://anonymous.4open.science/r/tooldec_rebuttal/), including `fsm.py` and `argument_fsm.py`.
2. Per Reviewer ydwb's request, We expanded the methodology section (**Section 3.3, Appendix A.6**) to explain how ToolDec enables generalization to unseen tools. We also included some new results from an ablation study on the robustness of LLM name writing in the case that good tool names are not available in **Appendix A.8**. Generalization is an important part of our contribution and we kindly ask you to consider it when evaluating our paper.
3. Per Reviewer R8ud's request, we made further clarifications about our experiment settings in **Section 4.3**. We explained more clearly the fairness of our comparison and gave more examples of ToolDec being more efficient than ToolDec in **Appendix A.5**.
4. Per Reviewer 7DQS's request, we ran experiments to evaluate the effectiveness of the mode-switching mechanism in ToolDec and reported the results in **Appendix A.7**.

Since the author-reviewer discussion period is closing, we kindly ask the reviewers who haven't responded to engage in the discussion. We are more than happy to hear your feedback and answer your further questions.

---

### Meta-Review · Area_Chair_Hhyt · 2023-12-11

**Metareview:**

The paper presents a constrained decoding method for LLMs that enables the use of external tools while minimizing syntax errors. This method involves restricting the model to decode exclusively from a predetermined set of valid tokens that align with the tool signatures. It is designed to be compatible with current LLM tool-use approaches, such as in-context learning and fine-tuning. Empirical evidence demonstrates that this method effectively eliminates syntax errors, leading to enhanced performance.

Several reviewers expressed concerns about the proposed method's ability to generalize to new and unseen tools. The authors responded by stating that the current generalization capability of ToolDec relies on the assumption of proper tool naming. After carefully considering the rebuttal and discussions, I recommend rejecting the paper due to its limited potential impact on generalization capability.

**Justification For Why Not Higher Score:**

As mentioned above, several reviewers expressed concerns about the proposed method's ability to generalize to new and unseen tools. The authors responded by stating that the current generalization capability of ToolDec relies on the assumption of proper tool naming. After carefully considering the rebuttal and discussions, I recommend rejecting the paper due to its limited potential impact on generalization capability.

**Justification For Why Not Lower Score:**

N/A

---

### Decision · Program_Chairs · 2024-01-16

Reject